# Behavioral and Psychological Symptoms and Associated Factors in Community-Dwelling Persons at the First Time of Dementia Diagnosis

**DOI:** 10.3390/ijerph19137765

**Published:** 2022-06-24

**Authors:** Gijung Jung, Jia Lee

**Affiliations:** 1Graduate School, Kyung Hee University, Seoul 02447, Korea; gijung1004@naver.com; 2College of Nursing Science, Kyung Hee University, Seoul 02447, Korea

**Keywords:** dementia, community-dwelling, behavioral symptoms, management

## Abstract

Background: Community-dwelling residents at potential risk of dementia and their families have difficulty detecting symptoms of dementia during an outbreak of coronavirus disease-19 (COVID-19). We explored the characteristics of behavioral and psychological symptoms of dementia (BPSD) in community-dwelling persons at the first time of dementia diagnosis and identified their associated variables. Methods: A cross-sectional study using secondary data of dementia diagnosis tests was conducted. Data were reported by professional nurses and clinicians from 355 persons at the first time of dementia diagnosis in South Korea. BPSD and their associated variables were measured with the Neuropsychiatric Inventory, the Korean version of the Consortium to Establish a Registry for Alzheimer’s Disease (CERAD-K) assessment handbook and electronic medical records. Results: The most common symptoms were apathy/indifference (72.1%), followed by irritability/lability (42.8%) and depression/dysphoria (42.0%). Hierarchical regression analyses showed that the strongest factor associated with BPSD was dementia type (β = −0.18, *p* = 0.001) mostly severer in frontotemporal dementia, followed by activities of daily living dependency (β = 0.15, *p* = 0.033), and number of medications (β = 0.10, *p* = 0.048). Conclusion: Providing information based on the study findings to families who are caring for persons at potential risk of dementia, may be able to detect dementia symptoms early and manage appropriate care.

## 1. Background

More than 886,000 South Koreans aged 65 years or older had dementia as of 2021, and the number is rapidly increasing with the aging population [1]. Behavioral and psychological symptoms of dementia (BPSD) such as aggression, wandering, irritation, etc. occur in all types of dementia including Alzheimer’s disease, vascular dementia, frontotemporal dementia and Lewy body dementia [2,3,4]. The prevalence of BPSD among community-dwellers ranged from 26 to 32% for new onset [5,6]. Moreover, continuous social distancing due to an outbreak of coronavirus disease-19 (COVID-19) induces a rapid increase in BPSD in approximately 60% of patients living in the community [6]. These BPSD not only threaten patients’ safety [7] and degrade their quality of life [8], but they also lead to a burden for family caregivers [9,10]. Therefore, support is necessary for individuals at potential risk of dementia living in the community and their families to understand the characteristics of BPSD and promote early identify symptoms.

Prior studies of BPSD were mainly conducted with persons with dementia who were already receiving treatment at geriatric hospitals or long-term care facilities while there have been few studies on community-dwelling persons at the first time of dementia diagnosis. A study of 293 community-dwelling persons with dementia proposed hearing, activities of daily living (ADL), cognitive function, and brain damage as factors affecting BPSD [11] while secondary analysis of the online system data of a dementia center suggested ADL only was related to BPSD [12]. In a study that interviewed dementia patients at home, cognitive function, ADL, and education level were factors affecting BPSD [13], while another study suggested that caregivers’ burden and stress exacerbated the BPSD [14].

However, these studies mainly examined examples in which a considerable amount of time had elapsed since diagnosis. When older adults and their families fail to detect BPSD at its early stages or are left without information on the causes for such symptoms, it will not only delay the diagnosis timing but also will adversely affect the quality of their lives [15,16]. Therefore, it is necessary to identify BPSD characteristics experienced by community-dwelling persons and their families when they first receive a dementia diagnosis and to investigate the factors affecting said initial symptoms.

## 2. Methods

### 2.1. Study Design and Participants

A cross-sectional study design using dementia diagnosis test data was employed. Participants were those who were diagnosed with dementia for the first time at the outpatient dementia clinics from a national hospital. They were diagnosed with dementia through diagnostic tests such as clinical evaluations, neuropsychological tests, BPSD tests, and brain imaging tests. The inclusion criteria were community-dwelling persons who had undergone a dementia diagnostic test and had no past history of dementia diagnosis.

The sample size was calculated using G*Power 3.1.9.2 [17] as 305 participants with a medium effect size of 0.25 of ANOVA which is an analysis method that requires the largest sample size in this study, a significance level of 0.05, power of 0.95, and 5 groups (i.e., the variable, “relationship” with the largest number of groups, see Table 1). To account for missing items, we obtained data from 360 patients who met the inclusion criteria. After excluding five patients with missing data, the data from 355 participants were analyzed.

### 2.2. Measures

Cognitive function. Cognitive function was measured by the Korean version of Mini-Mental State Examination (MMSE-KC) included in the Korean version of the Consortium to Establish a Registry for Alzheimer’s Disease (CERAD-K) assessment handbook [18]. The MMSE-KC is a tool that standardized the MMSE [19]. The tool consists of 26 questions assessing orientation, language, calculation, memory, attention, comprehension, and judgment. Total scores range from 0 to 30 and lower scores indicate more severe degradation of cognitive function. Cronbach’s αs were 0.92 in the development study [18] and 0.94 in this study.

Dementia severity. Dementia severity was measured with the Clinical Dementia Rating (CDR) tool included in the CERAD-K assessment handbook [20,21]. The items include memory, orientation, judgment and problem solving, community affairs, home and hobbies, and personal care. The total CDR score is based on a scale of 0–3: no dementia (CDR = 0), questionable dementia (CDR = 0.5), MCI (CDR = 1), moderate cognitive impairment (CDR = 2), and severe cognitive impairment (CDR = 3). The inter-rater reliability was Kappa 0.86–1.00 in the development study [20]; in this study, the intra-class correlation (ICC) coefficient was 0.96.

ADL. ADL were measured by the Blessed Dementia Scale–ADL included in the CERAD-K assessment handbook [18], which is a modified version of the Dementia Scale [22]. It consists of eight items related to daily living and three items concerning diet, defecation, and clothing habits. The total score is the sum of the scores according to the scoring method of each item: 0 to 5 points when there is mild dementia, 6 to 11 points when there is moderate dementia, and 12 to 17 points when the severe dementia is observed. Cronbach’s αs were 0.93 in the development study [18] and 0.94 in this study.

BPSD. For BPSD, the Korean version of the Neuropsychiatric Inventory (NPI-K) [23], a translated version of the NPI [24], was used. This tool consists of 12 items, including delusion, hallucination, agitation/aggression, depression/dysphoria, anxiety, elation/euphoria, apathy/indifference, disinhibition, irritability/lability, aberrant motor behavior, night-time behavior, and appetite/eating disorder. The tool evaluates BPSD by frequency, severity, and the level of pain experienced by the caregiver owing to patients’ symptoms. The frequency of BPSD is assessed on a 4-point scale, from 1 point for rare (less than once a week) to 4 points for very frequent (more than once a day). Severity is assessed on a 3-point scale, from 1 point for minor (hardly painful to the patients) to 3 points for severe (severe pain and disturbance to the patient that cannot be overcome, despite the efforts of the caregiver). The level of BPSD ranged from 0 points to 144 points, calculated by multiplying frequency by severity for each item. The higher the total score, the more severe the overall degree of BPSD. Cronbach’s α was 0.85 in the development study [24] and 0.90 in this study.

General characteristics. General characteristics included age, sex, education level, dementia type, dementia family history, number of chronic diseases, number of medications, visual and hearing impairments, caregivers’ relationship with patients, caregivers’ contact frequency with patients, and living environment. These were included on CERAD-K and patients’ electronic medical records during the diagnostic examination.

### 2.3. Data Collection and Ethical Considerations

After approval from an Institutional Review Board (IRB) from the study hospital (no. *-1901-176-1007), we obtained permission to access dementia diagnosis test results and electronic medical records, and then the data were extracted, verified, and organized for three months. Main data were extracted from the CERAD-K assessment handbook and NPI-K test result data, evaluated by medical staff for community-dwelling persons at the time of the first diagnosis of dementia from the outpatient clinic from 2016 to 2018. General information such as medication history was obtained through medical records included in patients’ electronic medical records.

All documents related to study participants were used for research purposes only, and a study number was used to protect their personal information from being identified. Only the data necessary for study analysis were collected. The collected data were stored in a file cabinet with a double lock, and computer files were password protected and only accessible to the principal investigator. All procedures were conducted in accordance with the institution’s IRB guidelines, and all data will be discarded after reviewing the study report.

### 2.4. Statistical Analysis

Data were analyzed with SPSS-PC statistical software (version 27.0, SPSS, Inc., Chicago, IL, USA). Participants’ characteristics were analyzed using frequency, percentage, mean, and standard deviation. The difference between BPSD per participants’ characteristics was analyzed with *t*-tests, ANOVA, and Scheffѐ’s post-hoc test. The influence of factors related to BPSD was analyzed with hierarchical regression analyses, and the correlations between independent variables were analyzed with Pearson’s and Spearman’s correlation coefficients. Tool reliability was analyzed with Cronbach’s αs and ICC coefficients.

## 3. Results

Concerning BPSD frequency among 355 community-dwelling persons at the first time of dementia diagnosis, apathy/indifference was the most symptom (*n* = 256, 72.1%), followed by irritability/lability (42.8%), depression/dysphoria (42.0%), agitation/aggression (32.4%), anxiety (32.4%) and delusion (22.8%). Concerning BPSD severity, apathy/indifference was the highest (3.84 ± 3.48), followed by irritability/lability (1.51 ± 2.50), depression/dysphoria (1.32 ± 2.07), anxiety (1.23 ± 2.40), agitation/aggression (1.19 ± 2.20) and delusion (1.14 ± 2.72) (Table 1, Figure 1). 

BPSD was highest in those with frontotemporal dementia (31.56 ± 19.22), followed by Lewy body dementia (22.31 ± 17.33), vascular dementia (13.76 ± 9.98) and Alzheimer’s disease (12.78 ± 13.14) (F = 14.21, *p* < 0.001). Concerning ADL dependency, those that scored moderate dependency of 6–11 showed the highest BPSD score (19.94 ± 15.60), followed by the severe dependency of ≥12 (14.67 ± 5.68) and mild dependency of ≤5 (12.02 ± 12.14) but significantly higher than those that scored mild dependency only. BPSD was significantly lower in those with the CDR of 0.5 points as compared to their counterparts. Concerning medication, those taking ≥2 drugs had significantly higher BPSD than those taking one drug only (F = 5.29, *p* = 0.005) (Table 2).

Factors associated with BPSD were selected as the independent variables of the prediction model: dementia type (r = 0.34, *p* < 0.001), number of medications (r = 0.16, *p* = 0.003), cognitive function (r = −0.12, *p* = 0.021), ADL dependency (r = 0.24, *p* < 0.001), and dementia severity (r = 0.24, *p* < 0.001). By analyzing the normality of the selected variables, the skewness ranged from −0.35 to 2.14 and the kurtosis ranged from −0.18 to 6.35, meeting the conditions of absolute values < 3 and <10, respectively. Thus, this satisfied the assumption of a normal distribution. The correlations between independent variables ranged from −0.02 to 0.67, which was <0.80; the tolerance ranged from 0.36 to 0.97, which was ≥0.1; and the variance inflation factor ranged from 1.03 to 2.76, which was not >10. Thus, there was no multicollinearity problem. For the main variables, the residual standardized with a linear relationship was not greater than the absolute value of 3, and the Cook’s distance value did not exceed the absolute value of 1.0; thus, there was no singular value. In addition, the Durbin–Watson test value was 1.88, which is close to the reference value of ±2; thus, there was no autocorrelation, and the independence of the error term was satisfied. Therefore, the basic assumption of the regression model for performing regression analyses was satisfied.

The hierarchical regression analysis showed that dementia type and the number of medications, which were the demographic and health status variables introduced in step 1, were significant, and had an explanatory power of 6.6%. By introducing cognitive function in step 2, the explanatory power was significant at 7.4%, and dementia type, number of medications, and cognitive function were all significant. By introducing ADL dependency in step 3, the explanatory power was significant at 10.0%, and dementia type, number of medications, and ADL dependency were significant but cognitive function was not. Lastly, by adding dementia severity in step 4, the explanatory power was significant at 10.2%, and dementia type, number of medications, and ADL dependency were significant but cognitive function and dementia severity were not (Table 3).

## 4. Discussion

This study identified the characteristics of BPSD and its associated factors in 355 community-dwelling persons at the first time of dementia diagnosis. Compared to most studies targeting patients with dementia or nursing home residents who have already progressed, identifying the characteristics of BPSD at the time of diagnosis of dementia can contribute to early diagnosis and response. The level of BPSD (14.61 ± 14.06) in this study was slightly higher than the scores (14.30 ± 13.90) in 67 community-dwelling persons with dementia [25] while it was lower than the scores in 241 nursing home residents including an Alzheimer’s disease group (32.92 ± 23.13) and vascular dementia group (28.76 ± 27.76) [26]. That is, the BPSD level of community-dwelling persons in this study was relatively lower than residents with dementia who had been in the facility longer and whose dementia had already progressed.

The most common of the BPSD subcategories was apathy/indifference (72.1%) which was higher than those seen in a study of community-dwelling persons with dementia (70.1%) [24] and among 56 patients with severe dementia living in the community (62.5%) [27]. In the latter study, prevalence of aberrant motor behavior (53.6%), delusion (50.0%), and agitation/aggression (50.0%) followed apathy/indifference whereas, in the current study, irritability/lability (42.8%) and depression/dysphoria (42.0%) followed apathy/indifference. It may be because apathy is the symptom that is consistently high from the early stage of dementia, and the frequency of symptoms differed according to the progress of dementia [28]. As apathy is easy for families to overlook, by reflecting the current results in information provision programs for the early detection of dementia, it would be possible to diagnose dementia before the aggravation of the disease.

Regression analyses revealed that the impact of BPSD on participants was in the order of dementia type (β = −0.18, *p* = 0.001), ADL dependency (β = 0.15, *p* = 0.033), and number of medications (β = 0.10, *p* = 0.048). Concerning dementia type, frontotemporal dementia (31.56 ± 19.22) and Lewy body dementia (22.31 ± 17.33) showed significantly higher scores of BPSD than Alzheimer’s disease (12.78 ± 13.14) and vascular dementia (13.76 ± 9.98) which is similar to a study result of 107 consecutive dementia patients that the level of BPSD was higher in frontotemporal dementia and Lewy body dementia compared to Alzheimer’s disease and vascular dementia [29]. This is thought to be because frontotemporal dementia is the frontal cortex responsible for the behavioral symptoms and Lewy body dementia is characterized by psychiatric symptoms. Therefore, symptom management programs for family caregivers should differentiate by dementia type.

Higher functional dependency was associated with higher BPSD which is consistent with a study of outpatients with Alzheimer’s disease that BPSD such as anxiety and depression increased when ADL decreased [11]. Although the current participants had a minor degree of dependency, ADL problems were associated with BPSD such as irritability and depression. BPSD was highest at moderate dependency level (19.94 ± 15.60) and decreased at severe dependency level (14.67 ± 5.68) in this study. This means that a high level of care for safety is required because BPSD occurs the most when dementia patients with some degree of activity experience difficulties in ADL. Rather, the amount of care felt by the caregivers may be relatively low if the level of ADL is very low. Therefore, efforts to promote patients’ independence such as support with assistive devices and ADL maintenance programs will reduce the incidence of BPSD at the early stage.

Most of the participants (70.4%) took at least one form of medication. BPSD was lowest among participants taking one medication (11.88 ± 11.88) and highest when they were taking three or more medications (18.16 ± 15.77), followed by two medications (17.62 ± 14.33). In a study that analyzed one-year changes in symptoms, anti-anxiety agents and sleeping pills were used most often, and increased medication use was associated with increased BPSD—a result that remained identical after one year [30]. Therefore, it is necessary to recognize that caution is warranted when prescribing and taking two or more medications. 

In this study, cognitive function and dementia severity did not significantly affect BPSD in the final hierarchical regression model. Concerning cognitive function, this is thought to be because of the introduction of ADL, which suppressed the effect of cognitive function on BPSD [31]. However, there was no significant difference in BPSD scores by group according to the level of cognitive function in ANOVA; and, considering that the lowest level of BPSD was observed in the minor cognitive impairment group rather than the group with normal cognitive function, it cannot be concluded that BPSD changed per the level of cognitive function with a consistent pattern. In a study analyzing cognitive impairment and BPSD in patients [32], delusion and hallucination were not related to the degree of cognitive impairment, and irritability and appetite showed a significant correlation. Therefore, when planning nursing interventions to improve cognitive function to alleviate BPSD, identifying the types of BPSD that are significantly correlated with cognitive function should be performed first.

There was a significant difference in the level of BPSD according to the severity; however, the influence was offset by the introduction of dementia severity in hierarchical regression analyses. It is thought that, since the dementia severity was significantly correlated with ADL dependency, the impact of dementia severity on BPSD was suppressed by ADL dependency. There is a study that the frequency of BPSD increased as the severity of dementia increased [12], while the other study that there were no significant changes according to the severity of dementia in delusion, depression and anxiety [33].

Therefore, when planning a BPSD management program for community-dwelling persons at the first time of dementia diagnosis and families caring for them, various aspects such as dementia type, ADL dependency and number and type of medications as well as individuals’ characteristics should be considered to predict and manage the occurrence of symptoms. Providing information on BPSD based on this study will contribute to the early diagnosis of dementia for community-dwelling persons and families. In addition, for the prevention and management of BPSD among this population, additional programs that aim to improve ADL should be disseminated to foster patients’ independence. The study findings may improve the wellbeing of community dwellers living with dementia and their families for the post-COVID new normal.

This study had some limitations. First, we need to be careful in generalizing the findings since we analyzed data collected from one national hospital. However, the hospital is very large with over 1700 beds and treats patients from all over the country. Moreover, unlike patients in dementia clinics or nursing homes, this study will help persons living alone or with their families in the community to understand the early symptoms of dementia and receive an early diagnosis. Second, as this was a secondary data analysis, the selection of variables was limited and they explained only 12% of the variance of the BPSD; thus, it is necessary to expand the scope of this research by conducting a prospective study that includes additional variables. Third, although it is difficult to repeatedly measure BPSD by continuously following up with those with dementia, it is necessary to investigate BPSD and the associated factors during the course of the disease through repeated measurements in a longitudinal study.

## 5. Conclusions

Among community-dwelling residents, the most common symptom of dementia at the time of diagnosis is apathy/indifference which is easy for families to overlook, followed by irritability/lability and depression/dysphoria. This finding helps the residents and their families detect and understand the symptoms of dementia early and its characteristics. Therefore, it is necessary to provide the information to community-dwelling persons and their families so that the diagnosis of dementia is not delayed or undiagnosed.

In this study, BPSD appeared mostly severer in frontotemporal dementia, with a relatively higher level of ADL dependency and a higher number of medications. Providing information based on the study findings to families who are caring for persons at potential risk of dementia may be able to prevent the aggravation of the disease by detecting dementia symptoms early and providing appropriate treatment. When planning a BPSD intervention program, it is necessary to classify each dementia type by considering the damaged area of the brain such as in frontotemporal dementia and Lewy body dementia. In addition, since BPSD deteriorate as functional dependency increases, it is necessary to intensively develop an intervention strategy that maintains patients’ independence by improving ADL and promoting the rehabilitation of existing cognitive functions. Additionally, since increased usage of antipsychotic medications was associated with increased BPSD, special attention is required concerning patients’ prescriptions. 

## Figures and Tables

**Figure 1 ijerph-19-07765-f001:**
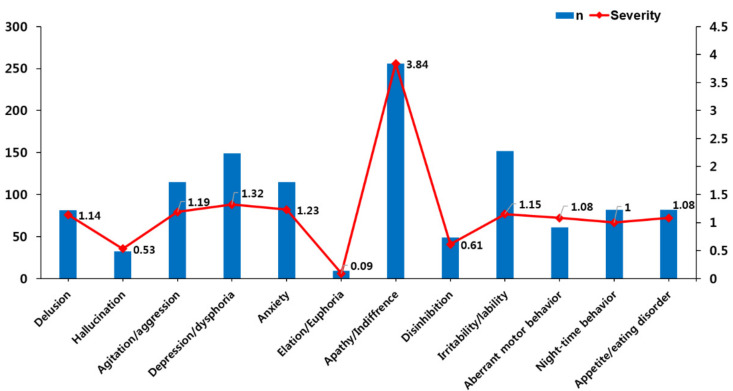
Prevalence and severity of behavioral and psychological symptoms at the first time of dementia diagnosis.

**Table 1 ijerph-19-07765-t001:** Participants’ Characteristics at the First Time of Dementia Diagnosis (*N* = 355).

Variable	Category	*n* (%)	*M* ± SD	Range
Age (years)	≤60 61–70 71–80 ≥81	47 (13.2) 97 (27.3) 153 (43.2) 58 (16.3)	71.88 ± 9.05	38–90
Sex	Male Female	143 (40.3) 212 (59.7)		
Education (years)	≤6 7–9 10–12 ≥13	144 (40.6) 39 (11.0) 82 (23.0) 90 (25.4)	9.42 ± 5.30	0–23
Dementia type	Alzheimer’s Vascular Lewy body Frontotemporal	245 (69.0) 66 (18.6) 26 (7.3) 18 (5.1)		
Family history of dementia	Yes No	69 (19.4) 286 (80.6)		
Number of comorbidities	0 1 2 ≥3	111 (31.3) 128 (36.0) 73 (20.6) 43 (12.1)	1.17 ± 1.07	0–5
Number of medications	0 1 2 ≥3	105 (29.6) 137 (38.6) 81 (22.8) 32 (9.0)	1.15 ± 1.02	0–4
Visual disturbances	Yes No	40 (11.3) 315 (88.7)		
Hearing disturbances	Yes No	20 (5.6) 335 (94.4)		
MMSE	≤9 10–19 20–23 ≥24	39 (10.9) 198 (55.7) 90 (25.3) 28 (7.9)	16.42 ± 5.67	0–30
ADL dependency	≤3 3.5–6 ≥6.5	124 (34.9) 158 (44.5) 73 (20.6)	4.57 ± 2.38	0–12.5
CDR	0.5 1 2	90 (25.4) 205 (57.7) 60 (16.9)		
Relationship	Spouse Daughter Son Daughter-in-law Relatives	175 (49.3) 93 (26.2) 61 (17.2) 15 (4.2) 11 (3.1)		
Frequency of contact (days/week)	≤2 3–4 ≥5	51 (14.4) 24 (6.7) 280 (78.9)		
BPSD			14.61 ± 14.06	1–76

M = mean; SD = standard deviation; ADL = activities of daily living; MMSE = Mini-Mental State Examination; CDR = clinical dementia rating; BPSD = behavioral and psychological symptoms of dementia.

**Table 2 ijerph-19-07765-t002:** Differences in BPSD per Participants’ Characteristics at the First Time of Dementia Diagnosis (*N* = 355).

Variable	Category	*n* (%)	*M ±* SD	*t* or *F* (*p*)	Scheffѐ’s Test
Age (years)	≤60 61–70 71–80 ≥81	47 (13.2) 97 (27.3) 153 (43.2) 58 (16.3)	15.19 ± 14.90 13.92 ± 13.53 14.12 ± 12.28 13.52 ± 13.74	0.15 (0.929)	
Sex	Male Female	143 (40.3) 212 (59.7)	14.59 ± 13.42 13.78 ± 13.04	0.57 (0.570)	
Education (years)	≤6 7–9 10–12 ≥13	144 (40.6) 39 (11.0) 82 (23.0) 90 (25.4)	14.72 ± 13.94 15.28 ± 15.99 12.04 ± 10.48 14.52 ± 12.83	0.91 (0.437)	
Dementia type	Alzheimer’s ^a^ Vascular ^b^ Lewy body ^c^ Frontotemporal ^d^	245 (69.0) 66 (18.6) 26 (7.3) 18 (4.5)	12.78 ± 13.14 13.76 ± 9.98 22.31 ± 17.33 31.56 ± 19.22	18.17 (<0.001)	a, b < c, d
Family history of dementia	Yes No	69 (19.4 )286 (80.6)	15.68 ± 15.47 13.73 ± 12.57	1.10 (0.271)	
Number of comorbidities	0 1 2 ≥3	111 (31.3) 128 (36.0) 73 (20.6) 43 (12.1)	14.59 ± 14.36 12.88 ± 10.98 14.66 ± 13.07 17.02 ± 16.69	0.65 (0.582)	
Number of medications	0 ^a^ 1 ^b^ 2 ^c^ ≥3 ^d^	105 (29.6) 137 (38.6) 81 (22.8) 32 (9.0)	13.09 ± 12.36 11.88 ± 11.88 17.62 ± 14.33 18.16 ± 15.77	4.57 (0.004)	b < c, d
Visual disturbances	Yes No	40 (11.3) 315 (88.7)	14.55 ± 13.31 14.55 ± 13.19	0.22 (0.823)	
Hearing disturbances	Yes No	20 (5.6) 335 (94.4)	16.10 ± 19.91 13.99 ± 12.70	0.47 (0.645)	
MMSE	≤9 10–19 20–23 ≥24	39 (10.9) 198 (55.7) 90 (25.3) 28 (7.9)	17.51 ± 12.24 14.45 ± 13.85 12.37 ± 11.38 14.11 ± 13.18	1.58 (0.195)	
ADL dependency	≤5 ^a^ 6–11 ^b^ ≥12 ^c^	254 (71.6) 91 (25.6) 10 (2.8)	12.02 ± 12.14 19.94 ± 15.60 14.67 ± 5.68	11.55 (<0.001)	a < b
CDR	0.5 ^a^ 1 ^b^ 2 ^c^	90 (25.4) 205 (57.7) 60 (16.9)	9.10 ± 9.11 15.12 ± 14.45 18.52 ± 11.61	10.92 (<0.001)	a < b, c
Relationship	Spouse Daughter Son Daughter-in-lawRelative	175 (49.3) 93 (26.2) 61 (17.2) 15 (4.2) 11 (3.1)	13.98 ± 13.96 14.02 ± 12.14 14.72 ± 13.59 12.93 ± 12.01 15.18 ± 9.59	0.85 (0.987)	
Frequency of contact (days/week)	≤2 3–4 ≥5	51 (14.4) 24 (6.7) 280 (78.9)	12.94 ± 12.02 11.46 ± 8.46 14.55 ± 13.70	0.84 (0.432)	

M = mean; SD = standard deviation; BPSD = behavioral and psychological symptoms of dementia; ADL = activities of daily living; CDR = clinical dementia rating.

**Table 3 ijerph-19-07765-t003:** Predictors of BPSD in Community-dwelling Persons at the First Time of Dementia Diagnosis (*n* = 355).

Variable	Model 1		Model 2		Model 3		Model 4	
	B	*β* (*p*)	B	*β* (*p*)	B	*β* (*p*)	B	*β* (*p*)
Dementia type ^†^	−6.19	−0.22 (<0.001)	−6.00	−0.21 (<0.001)	−5.29	−0.19 (<0.001)	−5.05	−0.18 (0.001)
Number of medications	1.67	0.13 (0.013)	1.67	0.13 (0.013)	1.44	0.11 (0.029)	1.32	0.10 (0.048)
MMSE			−0.25	−0.11 (0.037)	−0.02	−0.01 (0.902)	0.07	0.03 (0.662)
ADL dependency					1.09	0.20 (0.001)	0.82	0.15 (0.033)
CDR							2.13	0.10(0.162)
*R*² (△*R*²)	0.07		0.08 (0.01)		0.11 (0.03)		0.12 (0.01)	
Adj *R*²	0.07		0.07		0.10		0.10	
*F* (*p*)	13.43 (<0.001)		10.50 (<0.001)		10.81 (<0.001)		9.06 (<0.001)	

MMSE = Mini-Mental State Examination; ADL = activities of daily living; CDR = clinical dementia rating; ^†^ Dummy variables (Alzheimer’s disease = 1, other = 0).

## Data Availability

The datasets used and/or analyzed during the current study are available from the corresponding author on request.

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
