# Peer review of "Behavioral and Psychological Symptoms and Associated Factors in Community-Dwelling Persons at the First Time of Dementia Diagnosis"

_ijerph, 2022, doi:10.3390/ijerph19137765_

Round 1

Reviewer 1 Report

See attachment

Author Response

We truly appreciate your valuable comments. We made a table and used red font for the changes in the manuscript.

Reviewer 2 Report

This paper innovatively incorporates data from clinical diagnostic tests and from electronic medical records and identifies characteristics of BPSD and associated factors at the time of diagnosis of dementia among community dwellers in South Korea. It provides important implications for patients, caregivers, and BPSD interventions. Several improvements may further strengthen this paper.

1. This paper highlights the importance of early identification of BPSD and associated factors. I appreciate that the authors mentioned how the COVID-19 outbreak may affect diagnosis and characteristics of BPSD (page 1, line 10 & 32). In the discussion section, an explanation of how the findings may improve wellbeing of community dwellers living with dementia and their families during or after COVID will further strengthen the contribution of the study.

2. Providing the prevalence of BPSD among community dwellers may better contextualize the introduction (page 1, line 32). 

3. In the introduction, the authors identified a gap in the literature that is  lack of knowledge about BPSD characteristics and associated factors at the first time of dementia diagnosis (page 2, line 49). This may not be a big problem if community dwellers generally receive timely diagnoses and identifications of BPSD from physicians at the diagnosis of dementia. A brief discussion of issue(s) in early diagnosing BPSD among community dwellers may further improve the contribution of this paper. 

3. Clarify the time period of information extracted from electronic medical records (e.g., number of medications) - was this information collected on the day of dementia diagnosis (page 3, line 108)? 

4. In the discussion (Page 7, line 196), explain whether the length of stay in the facility and progression of dementia are the only differences between this study and previous ones. Clarify whether there are differences in measurements and samples across studies that may also contribute to differences in characteristics of BPSD. 

5. Provide potential explanations of why moderate ADL dependency instead of severe ADL dependency is associated with the highest BPSD (page 8, line 229).

6. Discuss how results from this study may differ from a nationally representative sample including both community dwellers and nursing home residents (page 9, line 267). 

7. Mention a limitation that this study does not identify a causal relationship between BPSD and its associated factors (page 9, line 275). 

8. Discuss whether delayed diagnosis or undiagnosis of dementia may affect the conclusion or the generalizability of the findings.  

Author Response

(The authors gave the same response as above.)
